# Orally Delivered Connexin43 Hemichannel Blocker, Tonabersat, Inhibits Vascular Breakdown and Inflammasome Activation in a Mouse Model of Diabetic Retinopathy

**DOI:** 10.3390/ijms24043876

**Published:** 2023-02-15

**Authors:** Odunayo O. Mugisho, Jyoti Aryal, Avik Shome, Heather Lyon, Monica L. Acosta, Colin R. Green, Ilva D. Rupenthal

**Affiliations:** 1Buchanan Ocular Therapeutics Unit, Department of Ophthalmology, University of Auckland, Auckland 1023, New Zealand; lola.mugisho@auckland.ac.nz (O.O.M.); jary684@aucklanduni.ac.nz (J.A.); a.shome@auckland.ac.nz (A.S.); heather-lyon@hotmail.co.uk (H.L.); i.rupenthal@auckland.ac.nz (I.D.R.); 2School of Optometry and Vision Science, University of Auckland, Auckland 1023, New Zealand; m.acosta@auckland.ac.nz; 3Department of Ophthalmology, University of Auckland, Auckland 1023, New Zealand

**Keywords:** retina, diabetic retinopathy, inflammasome, inflammation, vascular breakdown

## Abstract

Diabetic retinopathy (DR), a microvascular complication of diabetes, is associated with pronounced inflammation arising from the activation of a nucleotide-binding and oligomerization domain-like receptor (NLR) protein 3 (NLRP3) inflammasome. Cell culture models have shown that a connexin43 hemichannel blocker can prevent inflammasome activation in DR. The aim of this study was to evaluate the ocular safety and efficacy of tonabersat, an orally bioavailable connexin43 hemichannel blocker, to protect against DR signs in an inflammatory non-obese diabetic (NOD) DR mouse model. For retina safety studies, tonabersat was applied to retinal pigment epithelial (ARPE-19) cells or given orally to control NOD mice in the absence of any other stimuli. For efficacy studies, either tonabersat or a vehicle was given orally to the inflammatory NOD mouse model two hours before an intravitreal injection of pro-inflammatory cytokines, interleukin-1 beta, and tumour necrosis factor-alpha. Fundus and optical coherence tomography images were acquired at the baseline as well as at 2- and 7-day timepoints to assess microvascular abnormalities and sub-retinal fluid accumulation. Retinal inflammation and inflammasome activation were also assessed using immunohistochemistry. Tonabersat did not have any effect on ARPE-19 cells or control NOD mouse retinas in the absence of other stimuli. However, the tonabersat treatment in the inflammatory NOD mice significantly reduced macrovascular abnormalities, hyperreflective foci, sub-retinal fluid accumulation, vascular leak, inflammation, and inflammasome activation. These findings suggest that tonabersat may be a safe and effective treatment for DR.

## 1. Introduction

Diabetic retinopathy (DR), like most other complications of diabetes, is a chronic vascular disease associated with pronounced inflammation leading to the loss of vascular integrity. Many studies have associated the source of inflammation in DR with the activation of the nucleotide-binding and oligomerization domain-like receptor (NLR) protein 3 (NLRP3) inflammasome [1,2]. The NLRP3 inflammasome is a large protein complex with a tripartite structure, consisting of the NLRP3 protein, an adaptor apoptosis-associated speck-like protein containing a CARD (ASC) and caspase-1 [1,2]. The NLRP3 protein is an innate immunity receptor, which can induce inflammatory cascades when stimulated [1,2]. Its function to detect and respond to environmental irritants, endogenous danger signals, and pathogens is reflected in its association with a wide range of diseases, including infectious, autoinflammatory, and autoimmune disorders [1,2,3]. Several studies have shown that the NLRP3 inflammasome pathway is activated in DR patients compared to controls as well as in mouse models of advanced DR [4,5,6]. Moreover, a recent systematic review showed that the markers of inflammasome activation, interleukin 1-beta (IL-1β) and IL-18, increase in the vitreous and serum of patients with DR as the disease progresses [7]. Therefore, targeting the NLRP3 inflammasome could provide a therapeutic approach that acts upstream of anti-vascular endothelial growth factor (VEGF) agents.

Several studies have shown that blocking pathological connexin43 hemichannels is an effective way to prevent inflammasome activation and its perpetuation in DR [8,9,10,11,12,13]. In an in vitro study developed in retinal pigment epithelial (RPE) cells using a combination of high glucose and pro-inflammatory cytokines, IL-1β, and tumour necrosis factor-alpha (TNF-α), it was shown that blocking connexin43 hemichannels using Peptide5 inhibited ATP-dependent activation of the NLRP3 inflammasome [10]. In subsequent studies, Peptide5 was found to be effective in the inflammatory non-obese diabetic (NOD) mouse model in which the pro-inflammatory cytokines, IL-1β and TNF-α, were injected into the vitreous [11] to induce pronounced retinal glial cell activation, oedema, loss of vascular integrity, and neuronal death [14]. It was shown that blocking connexin43 hemichannels with Peptide5 not only prevented inflammasome activation, but also terminated the subsequent amplification and perpetuation of retinal inflammation [11], resulting in decreased vascular permeability and neovascularisation within the retina. However, the limitation of this study was that Peptide5 had to be delivered intravitreally, a route of administration that is both costly and inconvenient for patients. Another connexin43 hemichannel blocker, tonabersat, has been shown to inhibit ATP-dependent inflammasome activation in RPE cells exposed to high glucose and cytokines, as well as in a human donor retinal explant model of DR [15,16]. Tonabersat has the added advantage of being an orally bioavailable drug unlike Peptide5 which, similar to anti-VEGF agents, would necessitate intravitreal delivery in what is commonly a bilateral condition. Following oral administration, tonabersat is absorbed with a median t_max_ between 0.5 and 3 h and a long plasma half-life of 30–40 h (with no accumulation) enabling once daily dosing [17]. It has linear pharmacokinetics between 2 and 40 mg (in humans) with a proportional increase in the maximum serum concentration, and an increase in central nervous system concentrations (including the retina) in parallel with serum drug levels (that is, it crosses the blood–brain and blood–retinal barriers). There is no change in the elimination half-life with increasing doses and low inter-subject variability [17]. 

The aim of the present study is to determine the safety and efficacy of the orally bioavailable tonabersat to protect against inflammasome activation and signs of DR, including vascular breakdown, in the inflammatory NOD mouse model of DR. The NOD mouse model has been selected because it exhibits a heightened response in the presence of injury and has previously been used to evaluate the efficacy of connexin43 hemichannel blockers [11,13]. To evaluate drug safety, tonabersat was applied to RPE cells in the absence of other stimuli or given orally to NOD mice without an intravitreal pro-inflammatory cytokine injection. Markers of early inflammation and retinal structural changes were used to evaluate safety. The efficacy of tonabersat was examined by the administration of oral tonabersat 2 h before intravitreally introducing pro-inflammatory cytokines to the NOD mice. Fundus images were used to assess macrovascular abnormalities such as vessel dilation, beading, and tortuosity. From optical coherence tomography (OCT) images, retinal abnormalities, such as the formation of hyperreflective foci (HRFs), hypo-reflective areas indicative of subretinal fluid accumulation, and retinal layer thickness were determined. Finally, immunohistochemistry was used to examine molecular changes in the target protein (connexin43), blood vessel markers (plasmalemma vessel associated protein (PLVAP) and Isolectin-B4), general stress markers (glial fibrillary acidic protein (GFAP) and ionised calcium-binding adapter molecule 1 (Iba1)), and inflammasome specific markers (NLRP3 and cleaved caspase-1).

## 2. Results

### 2.1. Tonabersat Did Not Induce Any Changes in Uninjured ARPE-19 Cells or NOD Mice 

To determine the effect of a connexin43 hemichannel blocker, and therefore drug safety, in physiological conditions, cell viability was initially measured (Appendix A). The results showed that tonabersat did not reduce cell viability at any of the concentrations studied relative to the untreated conditions. To investigate whether a physiological connexin43 hemichannel blocker triggers inflammation, the release of pro-inflammatory cytokines and NF-κB localisation were assessed following the addition of tonabersat (Appendix A). The results showed that all measured cytokines were expressed in basal conditions with the tonabersat treatment decreasing the expression of IL-1β, IL-18, and VEGF but not IL-6, IL-8, and TNF-α. The immunohistochemical assessment of NF-κB localization showed that basally, NF-κB was expressed throughout the cell with no preference for the nucleus and remained unchanged after the addition tonabersat. 

To determine drug safety in the retina, tonabersat was administered orally to diabetic NOD mice. The results showed that in the absence of intravitreal pro-inflammatory cytokine stimulation, tonabersat on its own did not induce any retinal abnormalities (microaneurysms, haemorrhages, or ONH abnormalities) (Appendix A) or changes in the retinal layer thickness Appendix A). Furthermore, tonabersat did not induce any inflammatory processes as indicated by the lack of GFAP upregulation and ONL infiltration of Iba1-positive cells (Appendix A). 

### 2.2. Tonabersat Decreased the Incidence of Macrovascular Abnormalities

The results from the fundus images showed that the incidence α (Figure 1, Table 1), not the severity (Appendix A), of macrovascular abnormalities (vessel tortuosity, beading, and dilation) was affected by intravitreal pro-inflammatory cytokines and the tonabersat treatment. The results showed that tonabersat reduced the number of eyes with vessel dilation, beading, and tortuosity relative to the vehicle group (*p <* 0.0001 for all). Furthermore, relative to the vehicle group, tonabersat reduced the average number of tortuous vessels per eye (0.58 ± 1.00 vs. 6.50 ± 1.39, *p* < 0.0001) as well as the number of dilated vessels per eye (0.58 ± 1.51 vs. 3.25 ± 0.67, *p* = 0.0183). Although the tonabersat treatment (1.25 ± 0.42) appeared to reduce the average number of beaded vessels per eye relative to the vehicle group (3.13 ± 0.77), statistical significance was not reached (*p =* 0.1423). 

### 2.3. Tonabersat Prevented Retinal Hyperreflective Foci Formation, Swelling, and Subretinal Fluid Accumulation

The incidence of HRF seen in the ONL layer was determined from OCT images (Table 1 and the red arrows in Figure 2a). The results showed that tonabersat treatment significantly decreased the incidence of HRF which were only observed on day 2 relative to the vehicle group (*p <* 0.0001). To measure retinal swelling, the retinal layer thickness was determined from the OCT images (Figure 2b–d). Changes in the retinal layer thickness occurred on day 2 with the baseline levels restored by day 7. The results showed that the tonabersat treatment reduced cytokine-induced NFL-GCL-IPL (101.23 ± 5.57%, *p =* 0.0027), ONL (102.12 ± 7.36%, *p =* 0.0119), and total retina (98.33 ± 3.66%, *p* < 0.0001) thickness relative to the vehicle group (NFL-GCL-IPL: 113.38 ± 12.30%; ONL: 114.91 ± 12.60%; total retina: 112.22 ± 12.06%). The incidence of subretinal fluid accumulation on day 7 was also assessed from OCT images (Table 1 and the blue arrow in Figure 2a), with the tonabersat treatment decreasing the incidence of subretinal fluid accumulation relative to the vehicle group (*p <* 0.0001). 

### 2.4. Tonabersat Decreased the Number of PLVAP+ Vessels and the Number of Connexin43 Spots on PLVAP+ Vessels 

The retinal vessel integrity was determined by co-labelling mouse retina sections with PLVAP and Isolectin B4 (Figure 3a). The results showed that the tonabersat treatment (5.06 ± 0.64; *p* = 0.0417) decreased the number of PLVAP+ vessels relative to the vehicle group (9.00 ± 1.17) (Figure 3b). 

The tonabersat treatment also significantly decreased the number of connexin43 spots on PLVAP+ vessels in the OPL (34.88 ± 1.93; *p =* 0.0138) and ONL (21.44 ± 6.61; *p =* 0.0029) but not the NFL/GCL, INL, or IPL relative to the vehicle group (OPL:100.00 ± 19.58; ONL:100.00 ± 9.51) (Figure 4).

### 2.5. Tonabersat Inhibited GFAP Upregulation and Iba-1+ Cell ONL Infiltration

Retinal stress during inflammation was assessed using immunohistochemical analysis of the GFAP levels (Figure 5a,c) as well as ONL migration of the Iba-1+ cells (Figure 5b,d). The results showed that GFAP was expressed within the GCL in all experimental groups (Figure 5a). Tonabersat prevented GFAP upregulation within the inner retinal layers compared to the vehicle group (1.44 ± 0.23%; *p* = 0.0356) decreasing the overall area covered by GFAP labelling relative to the vehicle group (3.41 ± 0.77%) (Figure 5c). Tonabersat also prevented cytokine-induced migration of the Iba-1+ cells into the ONL such that the proportion of eyes with Iba-1+ cells in the INL was reduced from over 50% in the vehicle group (53.06 ± 7.08%) to around 11% in the tonabersat group (10.83 ± 4.90%, *p =* 0.0006) (Figure 5b,d). 

### 2.6. Tonabersat Inhibited NLRP3 and Cleaved Caspase-1 Upregulation

Inflammasome activation was assessed using antibodies against NLRP3 and cleaved caspase-1 (Figure 6 and Figure 7). Both NLRP3 and cleaved caspase-1 expression was observed throughout the retina in both groups, though the tonabersat treatment appeared to reduce the overall expression levels of both NLRP3 and cleaved caspase-1. Quantification revealed that the tonabersat treatment (79.67 ± 1.36%; *p =* 0.0359) significantly reduced the number of NLRP3 spots indicative of inflammasome complex assembly within the retina relative to the vehicle group (100.00 ± 4.26%) (Figure 6b). Specifically, tonabersat reduced NLRP3 spots within the ONL (53.63 ± 2.80%, *p =* 0.0167) relative to the vehicle group (100.00 ± 6.42%) (Figure 6c). While there was a trend towards reduced NLRP3 spots within the NFL-GCL (*p =* 0.7862) and IPL (*p =* 0.3441) with tonabersat treatment, this was not statistically significant. 

The tonabersat treatment (39.64 ± 1.20%, *p* = 0.0028) also decreased the number of cleaved caspase-1 spots, another marker of inflammasome complex assembly, within the retina compared to the vehicle group (100.00 ± 2.31%), especially within the IPL (20.09 ± 8.92% vs. 100.00 ± 2.86%, *p* = 0.0305), OPL (9.57 ± 2.43% vs. 100.00 ± 10.46%, *p* = 0.0137) and ONL (31.94 ± 16.67% vs. 100.00 ± 25.33%, *p =* 0.0457) (Figure 7b,c). There was a non-significant trend towards lower levels of cleaved caspase-1 within the NFL-GCL (*p =* 0.2838) and INL (*p =* 0.4273) with the tonabersat treatment. 

## 3. Discussion

The current gold standard treatment for DR, intravitreal anti-VEGF injections, is limited in two main ways [18,19,20,21]. Firstly, studies have revealed that while they may reduce vascular leaking by decreasing neovascularisation, they only target late-stage DR signs and do not alter disease progression. This is largely because anti-VEGF agents target downstream in the disease pathway without adequately addressing the underlying cause of the disease. Secondly, anti-VEGF agents are limited by their route of administration due to their large molecular weight and breakdown in the bloodstream. Intravitreal injections are uncomfortable and require specialist administration resulting in higher costs. As highlighted by Stewart et al. [22], drug regimens with fewer clinic visits and injections are becoming more favourable, especially when the desired visual outcomes are not compromised. As a result, other treatment regimens, preferably orally bioavailable therapies, are actively being sought. In this study, we describe the efficacy of an orally bioavailable treatment for DR, tonabersat, which targets an upstream disease pathway, the inflammasome pathway, to protect against vascular and molecular disease signs. 

Prior to evaluating the efficacy of tonabersat, its ocular safety was examined in uninjured ARPE-19 cells as well as in diabetic NOD mice (Appendix A). The results showed that increasing concentrations of tonabersat did not affect ARPE-19 cell viability, but instead lowered the levels of IL-1β, IL-18, and VEGF compared to basal. Furthermore, tonabersat addition did not induce any inflammatory response, evaluated using NF-κB in cells. In uninjured NOD mice, tonabersat administration did not affect the retinal microvasculature or retinal layer thickness. In addition, there were no signs of inflammation in mice who received tonabersat as evidenced by a lack of Müller cells or microglia activation. Taken together, these results suggest that tonabersat does not induce cell death or inflammation in basal conditions in vitro and in vivo and can therefore be considered safe. These findings are in line with previous studies in which tonabersat was found to be safe [17]. These findings are unsurprising given that tonabersat is believed to target connexin43 hemichannels to prevent ATP-mediated inflammasome activation. These hemichannels have been shown to open primarily under pathological conditions, normally forming gap junctions in physiological conditions before opening. Therefore, in the absence of pathological connexin43 hemichannels, tonabersat is expected to be safe at the dose used in the present study, though higher doses could potentially uncouple physiologically relevant gap junctions [23]. Nevertheless, the fact that tonabersat doses (20–80 mg in human participants is equivalent to 0.32–1.29 mg/kg in mice covering the 0.8 mg/kg used in the present study) have clinically been found to be safe and well tolerated [24] suggests that its supposed gap junction uncoupling actions [23] may be restricted to in vitro settings where clearance mechanisms are often non-existent. Furthermore, tonabersat has passed very high dose level toxicity and two-year carcinogenicity testing without any adverse effects [17]; this would not be possible if it was uncoupling gap junctions in vivo. 

Having established the ocular safety of tonabersat, the following studies examined its efficacy using the inflammatory NOD mouse model of DR. This model was created by an injection of pro-inflammatory cytokines, IL-1β and TNF-α, into the vitreous of diabetic NOD mice. While the acute nature of this model may limit its translatability to clinical practice for what is a chronic disease state, the model has previously been shown to mimic characteristic molecular and vascular DR signs present in the human condition [13]. In the present study, the results showed that oral tonabersat decreased the incidence of macrovascular DR signs; namely, vessel tortuosity and beading, and signs of active DR [25]. This idea was further supported by our finding that tonabersat protected against pro-inflammatory cytokine-mediated retinal layer thickening and significantly decreased the incidence of sub-retinal fluid accumulation. Sub-retinal fluid accumulation and retinal oedema, owing to the loss of endothelial cell integrity in an inflammatory environment, exacerbated by neovascularisation, are important clinical signs of DR that can occur at any stage of the disease [26,27,28]. As a result, anti-VEGF agents are currently used to decrease VEGF levels thereby reducing retinal neovascularisation in DR. However, while anti-VEGF targets both physiological and pathological VEGF resulting in off-target effects on otherwise healthy blood vessels, tonabersat targets a pathological pore (connexin43 hemichannels) upstream of VEGF, thereby shutting down the inflammatory pathways that lead to endothelial disruption and increased levels of VEGF and other inflammatory cytokines. This is evidenced by both its therapeutic effects as well as its safety, as discussed previously.

The vascular protection conferred by tonabersat was also demonstrated by its ability to decrease the number of PLVAP+ vessels in the retina. PLVAP is an endothelial cell specific protein that is absent in intact blood–retinal and blood–brain endothelia [29,30,31,32]. In pathological conditions such as DR, PLVAP has been shown to be significantly upregulated and this is associated with a loss of blood–retinal barrier integrity [33]. Wisniewska-Kruk et al. [34] previously showed that PLVAP plays a pro-angiogenic role in the retina and PLVAP inhibition both in vitro and in vivo decreased endothelial cell loss and neovascularisation. PLVAP levels were higher in the retina of Akimba mice, a model of advanced DR, with PLVAP levels correlating with the severity of fluorescein leakage [35]. Based on these studies and others, PLVAP is believed to be a marker of damaged vessels in the retina [35,36]. Our findings therefore suggest that PLVAP is expressed in the capillaries and venous vasculature in diabetic NOD mice and that tonabersat decreases the number of PLVAP+ cells in the retina, preventing blood vessel damage and protecting the blood–retinal barrier. 

The tonabersat treatment was also found to decrease retinal inflammation in this model. In recent years, it has been posited that chronic inflammation could be the instigator for DR development following years of hyperglycaemia [37,38,39,40]. In the present study, we found that tonabersat decreased GFAP expression suggesting decreased Müller cell activation, as well as ONL localisation of Iba-1+ cells suggesting decreased microglial activation and migration into the central retina. This is in line with previous studies that have shown, using in vitro and human organotypic retinal explant DR models, that tonabersat can protect against retinal inflammation [15,16]. Furthermore, in another study carried out in the same inflammatory NOD mouse model used here, we showed that blocking connexin43 hemichannels via intravitreal Peptide5 protected against Müller cell and microglial activation [11]. The effects of oral tonabersat on retinal inflammation has also been shown in other models of DR and AMD [9,41,42,43]. Mat Nor et al. [43] showed in a retinal light damage model of dry AMD as well as in a hyperglycaemic rat DR model that oral tonabersat treatment reduced the number of Iba-1+ cells and GFAP expression within retinal Müller cells. The protective effect of tonabersat on both Müller cells and microglia is particularly important to the pathogenesis of DR as both cell types play important roles in propagating and exacerbating inflammatory processes in the diseased retina. Müller cells are the main glial cells in the retina, forming structural support for nearby neurons and representing the connection between neuronal and vascular cells [44]. Müller cells have been shown to be susceptible to damage in DR with some studies suggesting that Müller cell activation could be one of the early markers of DR pathogenesis [45]. Importantly, Müller cells have been shown to secrete VEGF which contributes to inflammatory processes and vascular damage in the diabetic retina [46]. Furthermore, cross-talk between Müller cells and microglia has previously been shown to be an initiator of neuroinflammation in the retina [47]. It is thought that during the early stages of DR, Müller cells initiate retinal inflammation and then signal the involvement of microglia [48]. In that study, one of the pathways via which Müller cells are believed to recruit microglial participation is via ATP release, with Müller cells supposedly releasing ATP which in turn activates P2 × 7 receptors on microglia. Interestingly and as discussed below, tonabersat is thought to exert its actions by preventing connexin43-mediated ATP release and the resultant activation of the NLPR3 inflammasome. Taken together, tonabersat’s effect on Müller cell and microglial activation supports the idea that it acts on an upstream disease pathway and suggests that it could be effective in mitigating early DR signs. 

Several studies have associated the source of inflammation in DR with the activation of the NLRP3 inflammasome [4,5,6,10,12,15,49,50,51,52]. The NLRP3 inflammasome, the most common and most studied pathway of the innate immune system, has been shown to drive inflammatory processes in several chronic age- and non-age-related neuroinflammatory diseases [3,4,5,6,49,50,51,52,53,54,55,56,57,58,59,60,61,62,63,64,65,66,67,68,69,70,71,72,73,74,75,76,77,78,79,80,81,82,83,84,85]. In DR, the inflammasome pathway has been shown to be activated in DR patients [6,7], with secretion of the inflammasome markers, IL-1β and IL-18, being associated with faster disease progression [7]. Previously, we have shown that tonabersat can decrease retinal inflammation by targeting the NLRP3 inflammasome pathway in RPE cells exposed to high glucose and pro-inflammatory cytokines [15]. Tonabersat acts by reducing connexin43 hemichannel-mediated ATP release, thereby inhibiting ATP-mediated NLRP3 inflammasome activation [16]. In the present study, it was shown that tonabersat decreased NLRP3 and cleaved caspase-1 assembly into inflammasome complexes within the retina. Taken together, our findings and others support the idea that tonabersat, by blocking connexin43 hemichannel-mediated ATP release, decreases NLRP3 inflammasome activation and retinal inflammation in DR.

## 4. Materials and Methods

### 4.1. Animal Studies

Female NOD mice, between 15–17 weeks of age, were used in this study. There was no statistically significant difference between the mice in terms of weight, BMI, or blood glucose levels (Appendix A). The animals were born and housed in the Vernon Jansen Unit at the University of Auckland, New Zealand, under normal cyclic light conditions (12 h light: 12 h dark) and had access to food and water *ad libitum*. All animal experiments were in accordance with the Association for Research in Vision and Ophthalmology (ARVO) guidelines and were approved by the University of Auckland Animal Ethics Committee [AEC001787]. In vivo ocular assessments were conducted in mice anaesthetized by an intraperitoneal injection of ketamine (50 mg/kg; PhoenixPharm, Auckland, New Zealand) and medetomidine (0.5 mg/kg; Domitor^®^, Zoetis, Auckland, New Zealand). Following assessments, mice were awakened by an intraperitoneal injection of atipamezole (5 mg/kg; Antisedan^®^, Zoetis, Auckland, New Zealand). For the safety study, mice were randomly allocated to either an intravitreal saline injection only group (4 mice) or to a group fed with 0.8 mg/kg tonabersat by oral gavage and injected intravitreally with saline after 2 h (3 mice). For the efficacy study, mice were randomly allocated into a group receiving intravitreal cytokines only (vehicle group; 6 mice) or oral tonabersat (0.8 mg/kg) followed by an intravitreal cytokine injection after 2 h (6 mice). The number of mice required to achieve 80% statistical power was determined from previous experiments (11).

### 4.2. Intravitreal Injection of Pro-Inflammatory Cytokines

Mouse recombinant IL-1β (#RMIL1BI, Thermo Fisher Scientific, Auckland, New Zealand) and mouse recombinant TNF-α (RMTNFAI, Thermo Fisher Scientific, Auckland, New Zealand) were intravitreally injected together into the eyes of diabetic NOD mice to induce DR, as previously described [13]. Briefly, a solution containing 1 µL of each cytokine at a concentration of 500 ng/mL was injected into the temporal side immediately posterior to the limbus at the corneal–scleral intersection using a 10-µL Hamilton syringe with a 30G, 0.5 inch needle. Injections were performed under a dissection microscope to clearly visualize the needle and avoid damage to the cornea and lens. 

### 4.3. Tonabersat Treatment

Tonabersat (0.8 mg/kg; MedChemExpress, Princeton, NJ, USA) was delivered via oral gavage prior to ocular assessments for the safety study. This dose had been optimised in previous studies [9,43]. For the efficacy study, tonabersat was delivered 2 h before administration of intravitreal cytokines, in line with previous clinical trial data suggesting that the t_max_ of the drug can be achieved between 0.5 and 3 h [24]. Tonabersat was freshly prepared as a suspension by adding the drug to tap water and sonicating for 10 s every minute for 5 min. The vehicle group were given the equivalent volume of tap water. 

### 4.4. Funduscopy and Optical Coherence Tomography (OCT) Imaging

Following anaesthesia, mouse pupils were dilated with 1% tropicamide (Minims, Newbury, UK), then the cornea was kept moist using a lubricating eye gel (GenTeal^®^; Alcon, Zug, Switzerland). The Micron IV imaging system (Phoenix Research Labs, Pleasanton, CA, USA) was then used to acquire fundus and image-guided OCT images at the baseline (day 0), as well as 2- and 7-days post-treatment. Changes in retinal vasculature in terms of vessel beading, dilation, and tortuosity (Appendix A) were evaluated from fundus images on day 2 as previously described [13]. Vessel tortuosity, beading, and dilation were defined as an increase in the number of turns within a vessel, the presence of an inconsistent vascular tone, and an increase in the vessel diameter, respectively, compared to day 0. The proportion of eyes and the average number of blood vessels with each vascular abnormality were recorded. Vessel dilation was quantified by measuring the diameter of each vessel in each eye. Since vessel dilation is a pathology most prominently observed on day 2, the measurement was taken as a percentage change in vessel dilation on day 2 compared to day 0. The ‘draw line’ and ‘measure’ functions in the ImageJ software version 1.50i were used. Vessel tortuosity was quantified by measuring the perimeter of each vessel, as tortuous vessels tend to have a longer perimeter than non-tortuous vessels. Since vessel tortuosity change is a pathology observed on day 2, the measurement was taken as a percentage change in vessel tortuosity on day 2 compared to day 0. The ‘freehand tool’ was used to trace the vessel perimeter and the ‘measure’ function was used to determine the length in the ImageJ software version 1.50i. Vessel beading was quantified by counting the number of beads in each eye and calculating the average number of beads in each experimental group across the three time points studied.

From the OCT images, retinal abnormalities such as the presence of hyperreflective foci (HRF) indicative of microglia accumulation [86,87], hypo-reflective regions indicative of discrete areas of subretinal fluid accumulation, and the retinal layers’ thickness (Appendix A) were assessed as previously described [13]. As it was difficult to differentiate between the nerve fibre layer (NFL), ganglion cell layer (GCL), and inner plexiform layer (IPL) in mouse OCT images, these three layers were grouped together. The outer nuclear layer (ONL) and total retinal thickness were also quantified. The ‘draw line’ and ‘measure’ functions in the ImageJ software version 1.50i (National Institute of Health, USA) were used. Three measurements were taken across each of the three layers at the position of the optic nerve head (ONH). The baseline thickness was set to 100% and the result was expressed as a percentage change relative to the baseline on days 2 and 7.

### 4.5. Immunohistochemistry

Mouse eyes were enucleated following CO_2_ asphyxiation, then fixed in 4% paraformaldehyde in PBS for 1 h. Following two 10 min washes in 0.1 M PBS, tissues were passed through 10, 20, and 30% sucrose solutions before embedding in an optimal cutting temperature medium (Sakura, Alphen aan den Rijn, The Netherlands). Sagittal cryosections (14 µm) of the eye globe were mounted onto glass slides. The sections were washed in PBS thrice (10 min) before blocking for 1 h at room temperature with 10% normal goat or donkey serum and 0.1% TritonX-100 in PBS. The sections were subsequently incubated overnight at 4 °C with the primary antibodies as listed in Table 2. Following three 10 min washes the next day, the sections were incubated with the corresponding secondary antibody at room temperature for 2 h before the cell nuclei were stained using DAPI (1 µg/mL; D9542; Sigma-Aldrich, St. Louis, MO, USA). The sections were washed and mounted using an anti-fade reagent (Citifluor^TM^, Hatfield, PA, USA) and coverslips were sealed with nail polish. 

All images were taken with an Olympus FV1000 confocal laser scanning microscope (Olympus, Tokyo, Japan) and processed using the ImageJ software. Six images were taken per marker and per eye. The researcher was masked during immunofluorescence imaging and quantification to reduce any bias. To quantify GFAP expression, images containing DAPI were initially used to select the retinal layers using the freehand drawing tool in ImageJ. GFAP images were converted into binary images and an equal threshold value was set using the highest intensity images to reduce the background. The percentage area covered by the GFAP+ labelling was then quantified within the pre-selected retinal layers using the ‘measure’ tool. Co-localization of PLVAP (green) and Isolectin-B4 (red) was quantified by manually counting the orange-coloured blood vessels. To quantify the connexin43 spots on PLVAP+ vessels, the region of interest was set outlining the PLVAP+ vessels only. The connexin43 spots were counted specifically on PLVAP+ regions of interest. 

To quantify NLRP3 and cleaved caspase-1 spots, images containing DAPI were initially used to select the retinal layers using the freehand drawing tool in ImageJ. To subtract the background, NLRP3 (or cleaved caspase-1) images were transformed using the gaussian blur function with an equal sigma (radius) value across all images. The transformed image was then subtracted from the non-transformed image, images were converted to binary, and an equal threshold value was set. The number of NLRP3 or cleaved caspase-1 spots within the pre-determined regions of interest was then quantified using the ‘analyse’ function and expressed as a percentage of the vehicle group.

### 4.6. Statistical Analysis

The data are presented as an arithmetic mean +SEM. Statistical comparisons between groups were performed using either Student’s *t*-test or a two-way ANOVA with Sidak’s multiple comparisons post-hoc test. The test used is specified in each figure legend. Data with *p* ≤ 0.05 were considered to indicate a statistically significant difference. All statistical analyses were performed using GraphPad Prism, version 9.0.2. 

## 5. Conclusions

In conclusion, the present study has shown that tonabersat, delivered orally in the inflammatory NOD mouse model of DR, can protect against both vascular and inflammatory signs of the disease. In the first instance, it was shown that the tonabersat treatment was safe in both ARPE-19 cells and in uninjured NOD mouse retinas. In the cytokine-induced NOD mouse model of DR, tonabersat prevented NLRP3 inflammasome activation decreasing retinal inflammation, macrovascular abnormalities, sub-retinal fluid accumulation, and tissue oedema. Oral administration of tonabersat has the potential to be a safe and effective treatment for DR. 

## Figures and Tables

**Figure 1 ijms-24-03876-f001:**
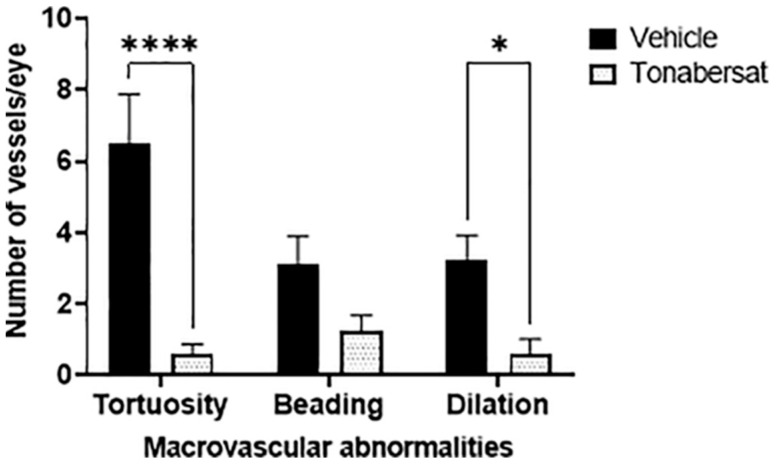
Tonabersat pre-treatment significantly reduced the formation of tortuous and dilated vessels and appeared to reduce the number of beaded vessels (although not reaching statistical significance for the latter). Data are presented as a mean + standard error of the mean (SEM). Statistical analyses were carried out using two-way ANOVA with Sidak’s multiple comparisons test. * *p* ≤ 0.05, **** *p* ≤ 0.0001. n = 8–12 eyes per group.

**Figure 2 ijms-24-03876-f002:**
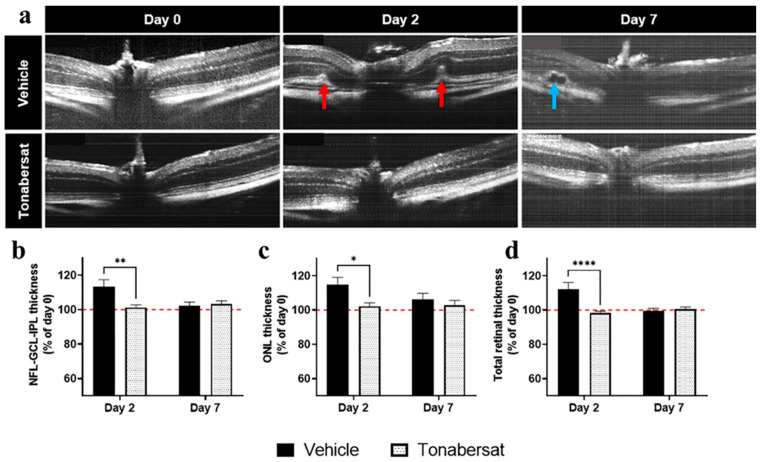
Tonabersat treatment decreased the incidence of macrovascular abnormalities, HRF formation, retinal swelling, and subretinal fluid accumulation relative to the vehicle group. (**a**) OCT images on days 0 (before treatment), 2, and 7. Red arrows indicate hyperreflective foci (HRF) on day 2 and the blue arrow an area of subretinal fluid accumulation on day 7. (**b**–**d**) On day 2, tonabersat treatment protected against cytokine-induced thickening of the (**b**) NFL-GCL-IPL, (**c**) ONL, and (**d**) the total retina relative to the vehicle group. Changes in retinal layer thickness were restored to baseline levels (red dotted line) by day 7. NFL = nerve fibre layer; GCL = ganglion cell layer; IPL = inner plexiform layer; ONL = outer nuclear layer. Statistical analyses were carried out using two-way ANOVA with Sidak’s multiple comparison’s test. * *p* ≤ 0.05, ** *p* ≤ 0.01, **** *p* ≤ 0.0001. n = 8–12 eyes per group.

**Figure 3 ijms-24-03876-f003:**
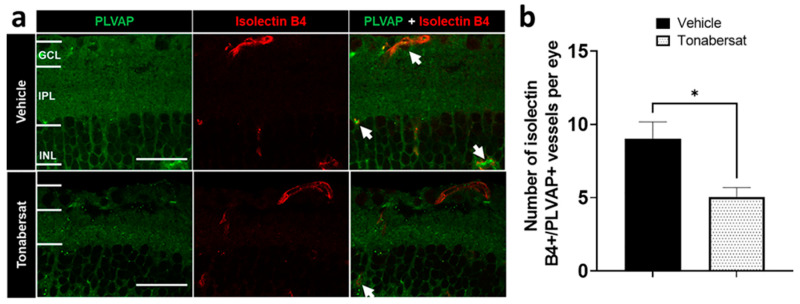
(**a**) Immunohistochemical images showing PLVAP (green) and Isolectin B4 (red) co-labelling in the vehicle and tonabersat groups. The arrows indicate examples of colocalization. GCL = ganglion cell layer; IPL = inner plexiform layer; INL = inner nuclear layer. (**b**) Tonabersat treatment decreased the number of PLVAP+/Isolectin B4+ blood vessels in the retina. Data are presented as a mean + SEM. Statistical analyses were carried out using Student’s *t*-test. * *p* ≤ 0.05. Scale bar = 50 µm. n = 8–12 eyes per group.

**Figure 4 ijms-24-03876-f004:**
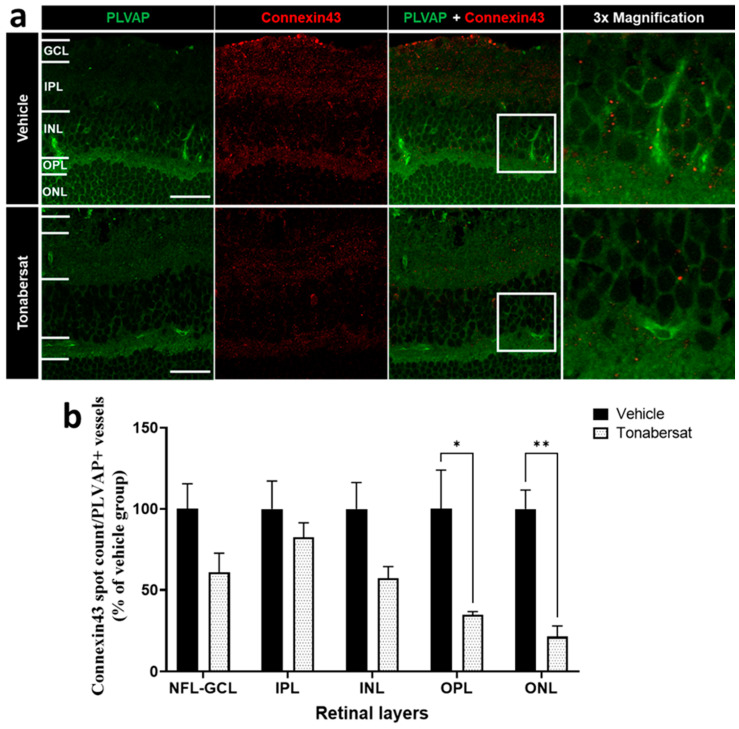
(**a**) Immunohistochemical images showing an example of PLVAP (green) and connexin43 (red) co-labelling. (**b**) Tonabersat treatment decreased the number of connexin43 spots on PLVAP+ vessels within the OPL and ONL but not the NFL-GCL, IPL, and INL. Data are presented as a mean + SEM. Statistical analyses were carried out using two-way ANOVA with Sidak’s multiple comparison’s test. * *p* ≤ 0.05, ** *p* ≤ 0.01. Scale bar = 50 µm. n = 8–12 eyes per group.

**Figure 5 ijms-24-03876-f005:**
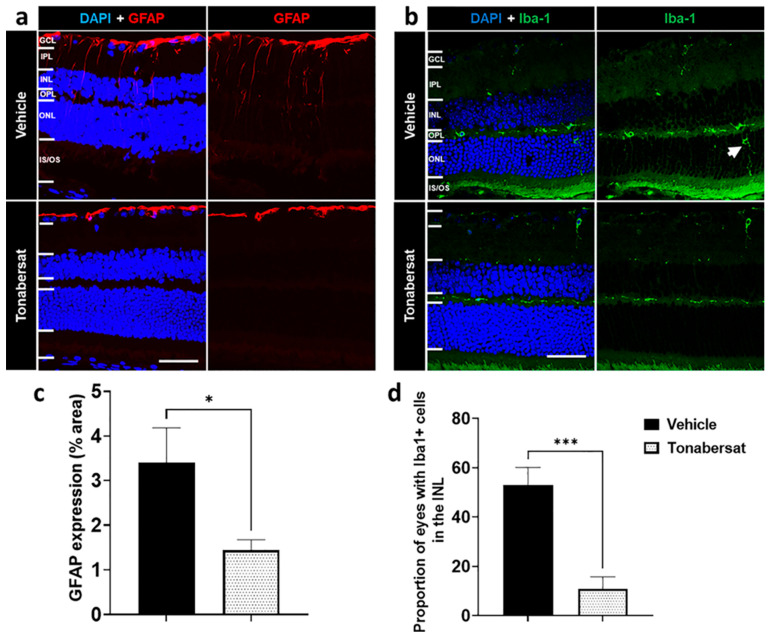
Immunohistochemical images showing (**a**) GFAP and (**b**) Iba-1 in vehicle and tonabersat-treated groups. (**a**) GFAP expression was restricted to the GCL in the tonabersat group but spanned from the GCL to the ONL in the vehicle group. (**b**) Vehicle alone led to migration of Iba-1+ cells into the ONL while tonabersat treatment restricted Iba-1+ cells to the IPL and OPL. The arrow indicates an Iba1 positive cell (activated microglia cell). Tonabersat treatment significantly decreased (**c**) overall GFAP levels and (**d**) the proportion of eyes with Iba-1+ cells in the INL. NFL = nerve fibre layer; GCL = ganglion cell layer; IPL = inner plexiform layer; INL = inner nuclear layer; OPL = outer plexiform layer; ONL = outer nuclear layer. Data are presented as a mean + SEM. Statistical analyses were carried out using a Student’s *t*-test. * *p* ≤ 0.05, *** *p* ≤ 0.001. Scale bar = 50 µm. n = 812 eyes per group.

**Figure 6 ijms-24-03876-f006:**
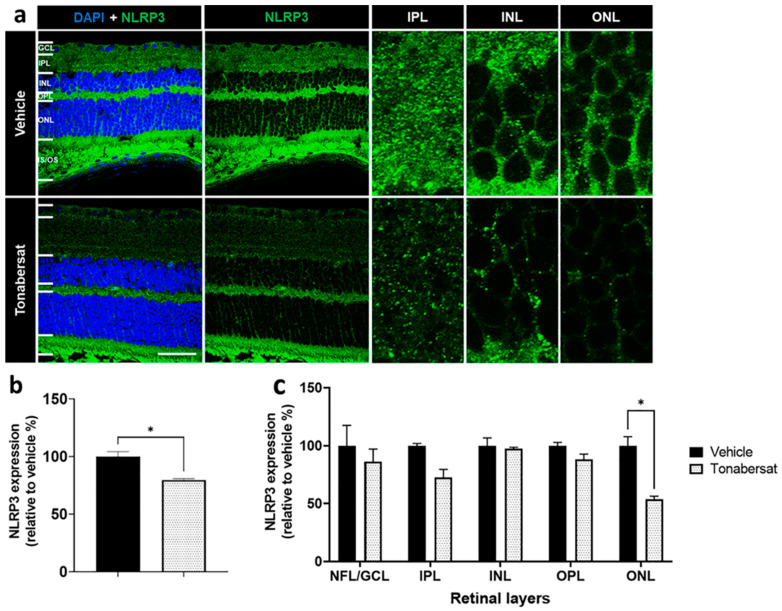
(**a**) NLRP3 expression was evident in both groups though overall levels appeared lower following tonabersat treatment compared to the vehicle group. GCL = ganglion cell layer; IPL = inner plexiform layer; INL = inner nuclear layer; OPL = outer plexiform layer; ONL = outer nuclear layer; IS/OS = inner and outer segments. Scale bar = 50 µm. (**b**) Tonabersat treatment significantly decreased overall NLRP3 levels relative to the vehicle group. Statistical analyses were carried out using a Student’s *t*-test. (**c**) Quantification of NLRP3 spots indicative of inflammasome complex assembly within specific retinal layers suggests that tonabersat reduced the number of NLRP3 spots significantly only in the ONL. Data are presented as a mean + SEM. Statistical analyses were carried out using two-way ANOVA with Sidak’s multiple comparison’s test. * *p* ≤ 0.05. n = 8–12 eyes per group.

**Figure 7 ijms-24-03876-f007:**
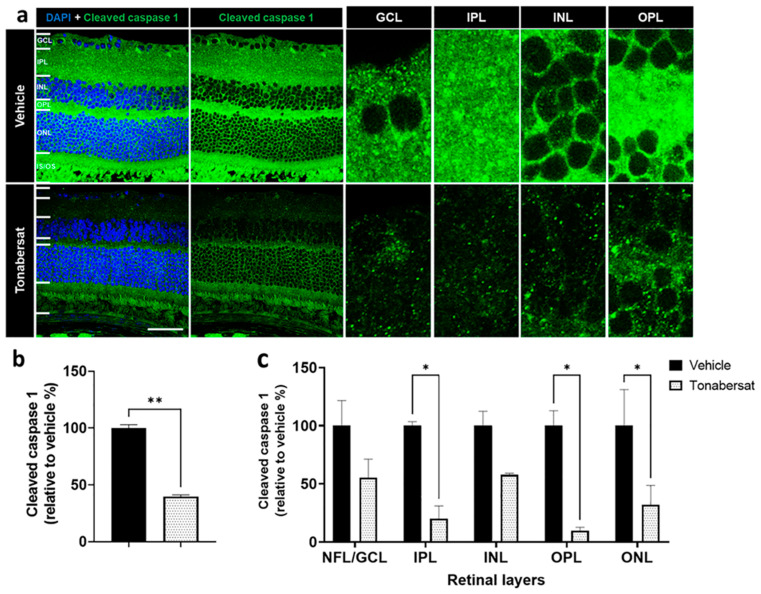
(**a**) Cleaved caspase-1 expression was evident in both groups though overall levels appeared lower following tonabersat treatment compared to the vehicle group. GCL = ganglion cell layer; IPL = inner plexiform layer; INL = inner nuclear layer; OPL = outer plexiform layer; ONL = outer nuclear layer; IS/OS = inner and outer segments. Scale bar = 50 µm. (**b**) Tonabersat treatment significantly decreased overall cleaved caspase-1 levels relative to the vehicle group. Statistical analyses were carried out using Student’s *t*-test. (**c**) Quantification of cleaved caspase-1 levels, another marker of inflammasome complex assembly, within specific retinal layers suggests that tonabersat treatment significantly reduced cleaved caspase-1 levels within the IPL, OPL, and ONL. Data are presented as a mean + SEM. Statistical analyses were carried out using two-way ANOVA with Sidak’s multiple comparison’s test. * *p* ≤ 0.05, ** *p* ≤ 0.01. n = 8–12 eyes per group.

**Table 1 ijms-24-03876-t001:** Incidence of vascular pathologies assessed from fundus and optical coherence tomography (OCT) images.

Pathology	Timepoint	Incidence per Treatment Group (Number of Animals and Percentage of Total Number)	
		Vehicle (n = 8)	Tonabersat (n = 12)
Vessel dilation	Day 2	8 (100%)	2 (16.7%) ****
Vessel tortuosity	Day 2	7 (87.5%)	4 (33.3%) ****
Vessel beading	Day 2	7 (87.5%)	6 (50%) ****
Hyperreflective foci	Day 2	3 (37.5%)	0 (0%) ****
Sub-retinal fluid accumulation	Day 7	2 (25%)	0 (0%) ****

Statistical analyses were carried out using the Fisher’s exact test. **** *p* < 0.0001.

**Table 2 ijms-24-03876-t002:** Primary and secondary antibodies used in this study.

Molecular Marker	Role	Antibody	Antibody Type	Working Dilution	Source
Connexin43	Target protein	Rabbit polyclonal	Primary	1:2000	Sigma Aldrich, St Louis, MO, USA #C6219
GFAP-Cy3	Müller cell and astrocyte marker	Mouse monoclonal	Primary	1:1000	Sigma Aldrich, #C9205
Iba1	Microglia marker	Rabbit monoclonal	Primary	1:2000	Abcam plc, Cambridge, UK #ab178846
NLRP3	Inflammasome marker	Goat polyclonal	Primary	1:500	Abcam plc, #ab4207
Cleaved caspase-1	Inflammasome marker	Rabbit polyclonal	Primary	1:50	Invitrogen, Auckland, New Zealand #PA5-38099
Isolectin-B4-A594	Blood vessel stain	Griffonia simplicifolia	Primary	1:100	Molecular Probes, Eugene, OR, USA #I-21413
PLVAP	Marker of leaky blood vessels	Mouse monoclonal	Primary	1:1000	Abcam plc, #ab27853
NF-κB	Inflammation marker	Rabbit polyclonal	Primary	1:1000	Abcam plc, #ab16502
Goat anti-rabbit Alexa Fluor 488	-	Goat polyclonal	Secondary	1:500	Invitrogen, #A11034
Donkey anti-rabbit Alexa Fluor 488	-	Donkey polyclonal	Secondary	1:500	Invitrogen, #A21206
Donkey anti-goat Cy3	-	Donkey polyclonal	Secondary	1:500	Jackson Immuno Research Laboratories Inc., Westgrove, PA, USA #705-165-147

## Data Availability

Data available on request.

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
