# Peer review of "Orally Delivered Connexin43 Hemichannel Blocker, Tonabersat, Inhibits Vascular Breakdown and Inflammasome Activation in a Mouse Model of Diabetic Retinopathy"

_ijms, 2023, doi:10.3390/ijms24043876_

Round 1
Reviewer 1 Report
I think that lines 374-376 in the discussion are not meant to be included.
Author Response
Reviewer 1
Lines 374 – 376 should not be included
We apologise for this error and have removed these sentences.
Reviewer 2 Report
Overview and general recommendation:
Investigating potential treatments for diabetic retinopathy is vital, as the number of patients diagnosed with diabetes is steadily increasing. DR is fast becoming a serious complication of diabetes, often leading to impaired vision and blindness in patients. This study is very interesting, in that it provides a clear mechanistic understanding of tonabersat and the potential use of this compound to treat DR. The current study is relevant and will be of interest to the journal’s readership. Overall, I found the paper relevant and easy to follow. The methodology was well-described and can be followed by subsequent researchers.
1. The authors described the study well and the period of experimentation provided compelling data, but the authors did not discuss any limitations of the study.
2. The small sample size should be taken into consideration when interpreting the results. The number of subjects in each group may be too small to come to any concrete conclusions, and a statement to this effect should be included.
3. Can the authors comment on the food given to the mice? Did the chow contribute to the study in any way? Was it inflammatory or anti-inflammatory? What was the macronutrient content (high fat, high carbohydrate)?
4. Can the authors comment on whether biochemical analysis was conducted on blood samples from the mice to observe inflammatory markers in blood, since diabetes is known for chronic inflammation?
The results are compelling at this stage and should be published after minor comments are addressed. Further research should be conducted in larger cohorts to validate this research.

Author Response
Reviewer 2
Investigating potential treatments for diabetic retinopathy is vital, as the number of patients diagnosed with diabetes is steadily increasing. DR is fast becoming a serious complication of diabetes, often leading to impaired vision and blindness in patients. This study is very interesting, in that it provides a clear mechanistic understanding of tonabersat and the potential use of this compound to treat DR. The current study is relevant and will be of interest to the journal’s readership. Overall, I found the paper relevant and easy to follow. The methodology was well-described and can be followed by subsequent researchers.
- The authors described the study well and the period of experimentation provided compelling data, but the authors did not discuss any limitations of the study.
We thank the reviewer for this constructive feedback. The main limitation in our study is the use of the inflammatory NOD mouse model which is an acute model that does not adequately reflect the chronicity of DR development in humans. However, it is important to note that chronicity of diseases such as DR are very difficult to replicate in mouse models and despite lacking this aspect, the inflammatory NOD mouse model has previously been shown to possess both the molecular and vascular signs of DR. To acknowledge this limitation in our study, we have included the following sentence.
While the acute nature of this model may limit its translatability to clinical practice for what is a chronic disease state, the model has previously been shown to mimic characteristic molecular and vascular DR signs present in the human condition [13].
- The small sample size should be taken into consideration when interpreting the results. The number of subjects in each group may be too small to come to any concrete conclusions, and a statement to this effect should be included.
In deciding the same size for our study, we conducted power calculations using the GFAP data from our previous Peptide5 study. We found that a sample size of 4 mice per group is sufficient to give a statistical power of 90%. In the present study using tonabersat, the power calculation from the GFAP data suggests that a sample size of 3 would give 90% power. Given that we used a sample size of 6 mice in the present study, we do not believe that the sample size is too small.
- Can the authors comment on the food given to the mice? Did the chow contribute to the study in any way? Was it inflammatory or anti-inflammatory? What was the macronutrient content (high fat, high carbohydrate)?
The mice were given standard chow which should be neither inflammatory nor anti-inflammatory. The same food was given to mice in both treatment groups and therefore the food composition should have no effect on the results obtained in the study. That is, any potential food related effects will be present in both the treatment and control arms.
- Can the authors comment on whether biochemical analysis was conducted on blood samples from the mice to observe inflammatory markers in blood, since diabetes is known for chronic inflammation?
Unfortunately, we did not conduct any biochemical analysis on the blood samples of the mice but acknowledge that this would have been interesting and relevant to the study. We are currently collecting mouse blood samples in a related study to investigate the effect of tonabersat treatment on serum inflammation markers in mice.
The results are compelling at this stage and should be published after minor comments are addressed. Further research should be conducted in larger cohorts to validate this research.
Reviewer 3 Report
Generally, the research is very well thought of, designed, and presented. The topic is very interesting and the results from this study are important in advancing treatment of retinopathy and other diabetes related complications.
There are few concerns that need to be addressed and these have been included in the attachment below.

Author Response
Reviewer 3
Comments to Authors
The authors aimed to evaluate ocular safety and efficacy of tonaberstat, an orally bioavailable connexin43 hemichannel blocker to protect against DR signs in an inflammatory non-obese diabetic mouse model. They used both animal models and cell culture to study the safety and efficacy of the drug. Generally, the article is very well written. The body of work is informative and important, especially the use of appropriate animal model to study the efficacy of tonaberstat. Nevertheless, there are few minor concerns that need to be addressed.
Minor criticisms
- The authors have used animal models and cell culture to determine the efficacy of tonabersat and used biomarkers and imaging studies to conclude safety of the drug. It is well known that biomarkers do not always translate to biological effect hence concluding safety by only measuring biomarkers and local effects could be misleading. The author should therefore provide evidence that the biomarkers of their choice truly reflect biological effects and translate to safety.
In preclinical studies, tonabersat has shown no significant toxicity in mice for up to 1000mg and no cardio, renal or respiratory effects at 2000mg/kg. Note that our study was at 0.8mg/kg. Long term treatments studies (Rat - 6 months treatment at doses up to 2000 mg/kg; Dog - 1 year at doses up to 200 mg/kg) showed no adverse effects and two-year carcinogenicity studies showed no carcinogenic effects. In clinical trials tonabersat has been taken by over 1500 human patients.
- A time course effect of tonabersat will add value to the study and conclusion.
If we understand the reviewer’s question correctly, we already have this information in our Introduction section. It was previously established in preclinical studies that Tonabersat has an oral Tmax absorption of 0.5 – 3h. Its plasma half-life is 30-40 h but there is no accumulation supporting a once daily dosing regimen. There was low inter-subject variability.
- Results discussed from line 100 to 129 whose figures are available online as supplementary figures are very important results. Some of these figures would add clarity and value to the findings if they appear on the main manuscript and not as supplementary figures.
In light of our comments above regarding preclinical and clinical safety data which establishes a clear tonabersat safety profile we do not agree that this is necessary. We think doing this will detract from the main thrust of the manuscript.
Reviewer 4 Report
Find attached

Author Response
The primary endpoint of this work is the safety of tonabersat, a connexin43 hemichannel blocker in mouse model of diabetic retinopathy while the secondary endpoint is efficacy in the same condition. The design of the work is good and the method is quite explicit. Nevertheless, there are few comments to improve the outlook of the work.
- Abstract Line 18- Efficacy comes before safety. So write efficacy and safety of ….
We have written this way because in the main manuscript, we described the safety results (section 2-1) before the efficacy studies. We hope the reviewer will agree with this.
- Line 19- NOD cannot mean two different things in abstract. Change one of them
Thank you for this observation. We have removed the NOD abbreviation in NLRP3 so that NOD in our manuscript only refers to the non-obese diabetic mouse.
- Line 22- Instead of vehicle, use control. Apply same to all places where vehicle appears in the work
We have referred to the control group as ‘vehicle’ because the mice were given equivalent volume of the ‘vehicle’ (tap water) while tonabersat suspension was given to the treated mice. We have clarified this in the methods section under ‘Tonabersat treatment’.
- Line 23- put a space between interleukin -1 and beta
The space has been inserted as suggested by the reviewer
- Introduction Line 53- remove “in which “
The words ‘in which’ have now been removed
- Line 58- Be consistent in the use of acronyms. NOD has been used for different meanings. First, it was in line 38 as “nucleotide-binding and oligomerization domain” and again in line 53 as “nonobese diabetic”. Be consistent
The changes mentioned in comment 2 above have been applied to the entire manuscript
- Line 64-66- The limitation of the study should be moved to the method section
We were referring here to the limitation of the previous work using Peptide5 to establish the need for the present study. We were not referring to the limitation of our present study in which we administered tonabersat not Peptide5.
- Line 81-97- Move line 81-97 to method section
- Results Line 100- The title of this subsection can be paraphrased as “Safety of Tonabersat on uninjured ARPE-19 cells or NOD mice”
We appreciate the reviewer’s comments for 8 and 9 but we think this is a stylistic preference, rather than a defect in the manuscript. Our preferred style is to me more direct in the subheadings so the reader knows at a glance what we have and can then read deeper as they prefer. We would not strongly object to these changes but prefer not to do so.
- Line 131- What does the error bar represent in Figure 1- SD or SEM. Please indicate. Indicate this in all the figures.
We apologize for the confusion. It was indicated in the methods section under ‘Statistical analysis’ that all data were presented as mean + SEM. To further clarify this, we have included this information in all figures as suggested by the reviewer.
- Line 119- instead of “Tonabersat decreased the incidence of macrovascular abnormalities”, I suggest “Correlation of Tonabersat treatment and macrovascular abnormalities”
- Line 139-140- Instead of “Tonabersat prevented pro-inflammatory cytokine-induced retinal hyperreflective foci formation, swelling, and subretinal fluid accumulation”, I suggest “Correlation of Tonabersat treatment and proinflammatory cytokine-induced retinal hyperreflective foci formation, swelling, and subretinal fluid accumulation”
Our response to 8 and 9 also applies here. If the editor insists we can make the change but we think the request states a stylistic preference rather than a criticism of the material in the manuscript.
- Line 153- Instead of using vehicle group, I suggest you use control groups
Please refer to our response to comment 3 made by this reviewer. Since animals were treated with the vehicle used to deliver the drug, it is a vehicle (or vehicle control) group rather than simply a control group.
- Line 165- Titles or subtitles should not be sentences. Make them as short and simple as possible. Apply same to all the subtitles.
Again, we respect the reviewer’s comments but his reflects a stylistic preference rather than a rule that should be applied. We prefer to retain what we have if the editor agrees.
- Line 215-216- Since there was a trend towards reduced NLRP3 spots within the NFL-GCL, it will be nice to indicate the p value. This is will show the nature of the trend. Do same in line 232-233 and anywhere else trends were used to present the result.
The p values have now been included thank you.
- Discussion Line 270- Insert the reference of the previous study showing safety
The reference has been added thank you.
- Conclusion It will good to have a section called Conclusion
We have now included a conclusion section as suggested by the reviewer. We agree that this strengthens the manuscript thank you.
- Line 373-376- This write up is confusing and should not be here.
This was left behind in error. We apologise - it has now been removed.